# Intranasal Delivery of Liposomes to Glioblastoma by Photostimulation of the Lymphatic System

**DOI:** 10.3390/pharmaceutics15010036

**Published:** 2022-12-22

**Authors:** Oxana Semyachkina-Glushkovskaya, Alexander Shirokov, Inna Blokhina, Valeria Telnova, Elena Vodovozova, Anna Alekseeva, Ivan Boldyrev, Ivan Fedosov, Alexander Dubrovsky, Alexandr Khorovodov, Andrey Terskov, Arina Evsukova, Daria Elovenko, Viktoria Adushkina, Maria Tzoy, Ilana Agranovich, Jürgen Kurths, Edik Rafailov

**Affiliations:** 1Institute of Physics, Humboldt University, Newtonstrasse 15, 12489 Berlin, Germany; 2Department of Biology, Saratov State University, Astrakhanskaya 82, 410012 Saratov, Russia; 3Institute of Biochemistry and Physiology of Plants and Microorganisms, Russian Academy of Sciences, Prospekt Entuziastov 13, 410049 Saratov, Russia; 4Shemyakin-Ovchinnikov Institute of Bioorganic Chemistry, Russian Academy of Sciences, Miklukho-Maklaya 16/10, 117997 Moscow, Russia; 5Department of Complexity Science, Potsdam Institute for Climate Impact Research, Telegrafenberg A31, 14473 Potsdam, Germany; 6Optoelectronics and Biomedical Photonics Group, AIPT, Aston University, Birmingham B4 7ET, UK

**Keywords:** liposomes, nasal-brain lymphatic system, intranasal drug delivery, photostimulation, glioblastoma

## Abstract

The blood–brain barrier (BBB) limits the delivery of majority of cancer drugs and thereby complicates brain tumor treatment. The nasal-brain-lymphatic system is discussed as a pathway for brain drug delivery overcoming the BBB. However, in most cases, this method is not sufficient to achieve a therapeutic effect due to brain drug delivery in a short distance. Therefore, it is necessary to develop technologies to overcome the obstacles facing nose-to-brain delivery of promising pharmaceuticals. In this study, we clearly demonstrate intranasal delivery of liposomes to the mouse brain reaching glioblastoma (GBM). In the experiments with ablation of the meningeal lymphatic network, we report an important role of meningeal pathway for intranasal delivery of liposomes to the brain. Our data revealed that GBM is characterized by a dramatic reduction of intranasal delivery of liposomes to the brain that was significantly improved by near-infrared (1267 nm) photostimulation of the lymphatic vessels in the area of the cribriform plate and the meninges. These results open new perspectives for non-invasive improvement of efficiency of intranasal delivery of cancer drugs to the brain tissues using nanocarriers and near-infrared laser-based therapeutic devices, which are commercially available and widely used in clinical practice.

## 1. Introduction

The blood–brain barrier (BBB) limits the delivery of most antitumor drugs and other pharmaceuticals into the brain [1,2,3,4]. However, the limitation of drug delivery into glioblastoma (GBM) via the BBB is controversial [5]. Indeed, GBM is characterized by disruption of the BBB [6,7]. Nevertheless, most clinical data suggest that around GBM the BBB is intact [5,8]. Therefore, the development of effective methods bypassing an intact BBB at the border of GBM is the first step in the GBM therapy [5].

The nasal-brain-lymphatic system is discussed as a pathway for brain drug delivery overcoming the BBB [9,10,11,12,13]. Intranasal delivery is based on the direct anatomic route between the olfactory neuroepithelium and the brain. However, in most cases, after intranasal administration of drugs, their concentration in the brain is not sufficient to achieve a therapeutic effect [9,10,11,12,13]. Therefore, it is necessary to develop technologies to overcome the obstacles facing nose-to-brain delivery of promising pharmaceuticals.

The cribriform plate is the main connective bridge between the brain fluids and the peripheral lymphatics [14,15,16,17,18,19]. The cerebral spinal fluid (CSF) flows through the sieve-like structure of the cribriform plate and in the space between the olfactory sensory nerve (OSN), where the CSF is absorbed by the nasal lymphatic vessels (LVs) with subsequent excretion into the cervical lymph nodes [16,20,21,22,23,24] (Figure 1b).

There has been speculation that the lymphatic system of the head, including the LVs of the cribriform plate, is a door from and into the brain, which maintains balance between the exit and the entrance of fluids and immune cells from and into the central nervous system (CNS) [25]. Recent studies have reported the meningeal “door” for immune cells, toxins, antigens, or wastes through the meningeal lymphatic vessels (MLVs) [25,26,27,28,29,30]. The MLVs might allow for the BBB bypass and to be an important pathway for drug delivery to GBM. Indeed, in our previous study, we clearly demonstrate effective delivery of liposomes to GBM via the MLVs after its administration in the deep cervical lymph nodes (dcLNs) and photostimulation [27]. Zhao et al. also report delivery of nanoparticles to the MLVs and the brain after its subcutaneous injection in the dcLNs [28]. These two pilot studies show the important role of the cervical lymphatic vessels in communication with the central nervous system (CNS). The results of these studies are an important informative platform for the development of innovative methods of lymphatic drug delivery to brain tissue, bypassing the BBB. However, the method of drug administration to the cervical lymph nodes is invasive and is not used in clinical practice, and therefore is not of clinical interest. Therefore, it is necessary to search for other non-invasive methods of lymphatic drug delivery to the brain tissue and directly to GBM. In this aspect, delivery of drugs through the nasal-brain-lymphatic system seems to be the most interesting since it can be easily introduced into clinical practice.

Photostimulation can be an innovative technology targeted for brain drug delivery via the nasal-brain lymphatic system. Transcranial photostimulation is a non-pharmacological and non-invasive therapy for numerous brain diseases [31,32,33,34,35,36,37]. There is strong evidence that photostimulation can regulate the relaxation and permeability of the LVs, activate the movement of immune cells in the lymph, and effectively manage lymphedema [37,38,39,40]. Recently, we demonstrated that near-infrared photo-effects (1267 nm) stimulates the clearance of different compounds from the brain via modulation of the lymphatic tone and contraction [38,39,40,41,42].

Liposomes are promising transporters to deliver anti-cancer drugs to GBM [27,43,44,45,46,47]. They offer numerous advantages, such as the similarity of the liposomal bilayer with the cell membrane; liposomes provide low drug toxicity, stability, and increased circulation time as well as targeting liposomes by modifying with antibodies, and polypeptides [43,48,49]. These properties of liposomes make them promising transporters to deliver anti-cancer drugs to the CNS and thus, to improve the GBM therapy [43,47,50]. Among different types of liposomes, those that incorporate with gangliosides are of greatest interest for brain drug delivery [47,51]. Gangliosides consist of a hydrophilic oligosaccharide with sialic acid residues and a hydrophobic ceramide moiety. They are widely synthetized in the neural tissue of mammalians and are involved in many physiological processes, including modulation of ion transport, differentiation of neurons, activation of the G protein-coupled receptors and the immune system [52,53]. There is evidence that liposomes containing monosialotetrahexosylganglioside (ganglioside GM_1_) can overcome the BBB [47,51]. Indeed, in our previous study, we demonstrated delivery of the GM_1_-liposomes to the mouse brain after photodynamic opening of the BBB [54].

In this work, we aim to study delivery of the GM_1_-liposomes to the mouse brain and fluorescent GBM via the nasal-brain lymphatic system, bypassing the BBB. We also hypothesized that near-infrared (1267 nm) photostimulation of the LVs located in the cribriform plate and the MLVs can improve intranasal delivery of liposomes to GBM.

## 2. Materials and Methods

### 2.1. Subjects

Male C57BL/6 mice (25–28 g) were used in all experiments and were obtained from the National Laboratory Animal Resource Centre in (Pushchino, Moscow, Russia). The animals were housed under standard laboratory conditions, with access to food and water, ad libitum. All experimental procedures were performed in accordance with the “Guide for the Care and Use of Laboratory Animals”, Directive 2010/63/EU on the Protection of Animals Used for Scientific Purposes, and the guidelines from the Ministry of Science and High Education of the Russian Federation (No. 742 from 13.11.1984), which have been approved by the Bioethics Commission of the Saratov State University (Protocol No. 7). The experiments were performed in the following experiment groups: (1) The confocal and multiphoton analysis of intranasal delivery of liposomes to the ethmoid bone, the meninges, and the brain, *n* = 7; (2–5)—the study of the role of the MLVs in intranasal delivery of liposomes to the brain using ablation of the MLVs (visudyne + laser) (2) and three controls groups, including sham mice (3), animals treated by visudyne (4) or laser (5) only, *n* = 5 in each group; (6 and 7)—the confocal analysis of intranasal delivery of liposomes to the brain in the healthy group (HG) and in the GBM group, respectively, *n* = 5 in each group; (8 and 9)—the study of photostimulation of intranasal delivery of liposomes to GBM in mice treated and not treated by infra-red laser (1267 nm), respectively, *n* = 7 in each group.

### 2.2. Synthesis of GM_1_-Liposomes

Fluorescently labelled GM_1_-liposomes were prepared as described earlier [54]. Briefly, a lipid film obtained from egg yolk phosphatidylcholine (Lipoid GmbH, Germany)—ganglioside GM_1_ from bovine brain (Sigma Chemical Co., St. Louis, MO, USA), 9:1 (by mol.), and 1 mol. % the fluorescent lipid probes BODIPY-phosphatidylcholine synthesized as described earlier [55] (λ_ex_ = 497 nm, λ_em_ = 505 nm) or Rh101-dioleoylglycerol (λ_ex_ = 578 nm, λ_em_ = 602 nm) synthesized by Prof. Julian Molotkovsky from the Shemyakin-Ovchinnikov Institute of Bioorganic Chemistry, RAS, was hydrated in physiological saline (phosphate buffer with 0.5 mM EDTA, pH 6.8; total lipid concentration 25 mM). The resulting suspension was subjected to seven cycles of freezing/thawing (liquid nitrogen/+40 °C), and extruded ten times through polycarbonate membrane filters (Nucleopore, Boston, MA, USA) with a 100 nm pore diameter using an Avanti Mini-extruder (Northern Lipids, Burnaby, BC, Canada). Particle size was measured by dynamic light scattering with the Brookhaven equipment (Brookhaven Instruments Corp. 90Plus Particle Sizing Software ver. 4.02, Holtsville, NY, USA) in at least three runs per sample: effective diameters and polydispersity indexes were 99.6 nm and 0.160 for green liposomes and 91.4 nm and 0.147 for red liposomes, respectively. Zeta potential of liposomes was measured using a ZetaPALS analyzer (Brookhaven Instruments Corp., Holtsville, NY, USA), was –48.3 ± 0.9 mV [55].

### 2.3. Implantation of EPNT-5-TagRFP GBM

Mice were deeply anesthetized with intraperitoneal Zoletil (Virbac, Carros, France) in a dose of 20 µg/kg and moved into a stereotaxic head holder and immobilized on the stereotactic system (Narishige, Tokyo, Japan) by fixation of the head. The scalp of the anesthetized mice was shaved and scrubbed with betadine three times followed by an alcohol rinse. Hair was removed at the site of the planned operation and a cut was made in the area of the planned injection. An incision was made over the sagittal crest from the bregma to the lambdoid suture and the periosteal membrane was removed. A small dental drill was used to create a burr hole through the bone without tearing the dura matter in the exposed cranium 1 mm anterior and 1 mm lateral to the bregma.

The cell line EPNT-5 of mouse GBM was obtained from the Russian Cell Culture Collection of Vertebrates, Institute of Cytology, Russian Academy of Sciences, (St. Petersburg, Russia). The cell line EPNT-5 of GBM was chosen for research because it has certain metabolic and structural characteristics close to those of a human GBM, in particular hyper-expression of α-glycerophosphate dehydrogenase enzyme and GABA-A channel subunit, cellular polymorphism, high genetic heterogeneity, which makes GBM resistant to therapy. The transfected EPNT-5-TagRFP cell line was obtained using the protocol of A. Shirokov et al. and used to study the growth of fluorescent GBM [56]. EPNT-5 cells were cultured in Dulbecco’s modified eagle medium (DMEN, PanEco, Moscow, Russia) containing 10% fetal calf serum (Biosera, Marikina, Philippines), 4 mM glutamine (PanEco, Moscow, Russia), penicillin (50 IU/mL), and streptomycin (50 mg/mL) (PanEco, Moscow, Russia). Versen’s solution (PanEco, Moscow, Russia) with the addition of 0.25% trypsin (Gibco, Thermo Fisher, Waltham, USA) was used to remove cells from the surface of the culture plastic. Cells were cultured in a humid environment in a CO_2_ incubator at a temperature of 37 °C, with a content of 5% CO_2_. The number of cells was counted on the TC20 Bio-Rad automatic cell counter, and the viability analysis was determined using a trypan blue dye. Mouse EPNT-5 of GBM cells were transfected with pTagRFP-C DNA plasmid using the method of liposomal transfection followed by selection using geneticin (G418 antibiotic, neomycin analogue) on the base research center “Symbiosis” IBPPM RAS (Saratov, Russia). The resulting cell line EPNT-5-TagRFP has stable cultural and morphological characteristics.

The GBM cells (5 × 10^5^ cells per mice) were injected at a depth of 1 mm from the brain surface with a Hamilton microsyringe in a volume of 7 µL at a rate of 0.5 µL/min. Thereafter, the burr hole was sealed with sterile bone wax and tissue glue and the wound sutured closed with 3–0 absorbable suture material. After implantation of the GBM cells, the wound was closed and treated with 2%-brilliant green solution. The mice were removed from the stereotaxic head holder, given 0.01 mg/kg buprenorphine, s.c. and 50K bicillin, i.m. and returned to their cages after recovery in the temperature-controlled recovery cage and moved back to animal facility after recovery. The animal was placed in a clean cage. The growth of fluorescent glioblastoma EPNT-5-TagRFP was assessed by confocal microscopy using the confocal laser scanning microscope Nikon A1R MP (Nikon Instruments Inc., Tokyo, Japan).

### 2.4. Immunohistochemistry (IHC)

For confocal imaging of delivery of liposomes to the tested tissues, we used the protocol for the IHC analysis with the markers for the lymphatic vessel endothelial hyaluronan receptor 1 (LYVE1) and for astrocytes by the glial fibrillary acidic protein (GFAP). For the IHC analysis, the brain, the meninges, and the dcLN tissues were collected and free-floating sections were prepared. The tested tissues were fixed for 48 h in a 4% saline solution-buffered formalin, then sections of tissues with a thickness of 40–50 microns were cut on a vibrotome (Leica, Wetzlar, Germany). The antigen expression was evaluated on sections of the mouse brain, the meninges and the dcLN according to the standard method of simultaneous combined staining of the drug (abcam protocols for free-floating sections) using a confocal laser scanning microscope Nikon A1R MP, Nikon Instruments Inc.).

The nonspecific activity was blocked by 2 h incubation at room temperature with 10% BSA in a solution of 0.2% Triton X-100 in PBS. Solubilization of cell membranes was carried out during 1 h incubation at room temperature in a solution of 1% Triton X-100 in PBS. Incubation with primary antibodies in a 1:500 dilution took place overnight at 4 °C: with rabbit anti- Lyve-1 antibody (1:500; ab 218535, Abcam, Cambridge, UK) and rabbit antibodies to GFAP (1:500; ab 207165; Abcam, Biomedical Campus Cambridge, Cambridge, UK). At all stages, the samples were washed 3–4 times with 5-min incubation in a washing solution. After that, the corresponding secondary antibodies goat anti—rabbit IgG (H+L) Alexa Four 405; goat anti—rabbit IgG (H+L) Alexa Flour 488; (Invitrogen, Molecular Samples, Eugene, Oregon, USA) were applied. At the final stage, the sections were transferred to the glass and 15 µL of mounting liquid (50% glycerin in PBS) was applied to the section. The preparation was covered with a cover glass and confocal microscopy was performed.

### 2.5. Confocal and Multiphoton Images

The ex vivo visualization of the distribution of fluorescent dyes in the brains, GBM, the meninges, the ethmoid bone, and the dcLNs was performed with an A1R MP confocal and the multiphoton microscopy system based on the Ni-E focusing nosepiece upright microscope (Nikon, Japan). Four laser excitation sources with 405 nm, 488 nm, 561 nm, 640 nm wavelengths and a set of photomultipliers as detectors were used for multichannel confocal imaging with 2× and 20× objectives.

Fresh samples of whole brain, the meninges, the ethmoid bone, and the dcLNs were immersed in saline solution in a Petri dish and covered with horizontally oriented cover glass. The Petri dish was placed on an electromechanical stage of the confocal imaging system.

IHC stained tissue sections were mounted on standard microscope slides and were imaged using a slide adapter fixed on the same stage.

The images were obtained using NIS-Elements software (Nikon Instruments Inc.) and analyzed using Fiji software (Open-source image processing software) and Vaa3D (Open Source visualization and analysis software).

In all experiments, the cerebral vessels were filled with the Evans Blue dye (2 mg∕100 g, 1% solution in physiological 0.9% saline, Sigma-Aldrich, St. Louis, MO, USA) that was injected intravenously 30 min before confocal and multiphoton microscopy via the tail vein.

The liposome (5 µL) was injected into the nasal cavity 3 h and Lyve-1 conjugated with Alexa Fluor 488 (5 µL, 0.1 µL/min, NBP1-43411AF488, Novus Biologicals, Centennial, CO, USA) was injected 90 min into the cisterna magna before confocal and multiphoton microscopy.

### 2.6. Quantitative Analysis of Confocal Images

For the quantitative analysis of intensity signal from liposomes in the brain and the meninges, GBM, the dcLNs, ImageJ was used for image data processing and analysis.

Lymphatic ablation and lymphatic regression were measured by the analysis of average fluorescent signal from the MLVs labeled by Alexa Flour 488 Lyve-1 antibody over the entire slice in the sham, the visudyne, the laser, and the visudyne + laser groups; the blood vessels were filled by Evans Blue and were distinguished in the red channel (640 nm).

### 2.7. Photo-Damages of the MLVs

To ablate the MLVs, visudyne treatment was carried out according to a previous publication [29]. Briefly, mice from the visudyne + laser group were anesthetized with ketamine hydrochloride and visudyne (APExBIO, Cat. No. A8327, 5 μL) was injected into the cisterna magna at a speed of 1 μL/min. Fifteen minutes later, with a 689-nm wavelength laser (Changchun Laser Technology, a dose of 50 J/cm^2^) was applied through the skull in different places, including the cisterna magna, the left and right transverse sinuses, the superior sagittal sinus, and the junction of all sinuses. For the control groups (sham and visudyne only), mice were injected (5 μL, into the cisterna magna) by physiological saline or visudyne, respectively; the laser (control) group included mice treated by laser (1267 nm) irradiation without visudyne. The eyes of mice were protected during photo-ablation of the MLVs. In all groups, 7 days after photo-ablation of the MLVs, liposomes were injected into the nasal cavity 3 h before removal of the meninges and confocal microscopy.

### 2.8. Laser Iradiation Scheme and Dose Calculation

A fiber Bragg grating wavelength locked high-power laser diode (LD-1267-FBG-350, Innolume, Dortmund, Germany) emitting at 1267 nm was used as a source of irradiation. The laser diode was pigtailed with a single-mode distal fiber ended by the collimation optics to provide a 5 mm beam diameter for photostimulation of the MLVs in the area of the transverse sinus and the superior sagittal sinus and a 1 mm beam diameter for photostimulation of the LVs located in the nasal cavity. The mice were recovered after the 7 days of GBM cells transplantation. Thereafter, the heads of mice were shaved and the scalp was removed, then they were fixed in a stereotaxic frame and immediately after intranasal administration of liposomes (5 µL) animals were irradiated with a near-infrared laser using the algorithm published earlier [27]: 17 min light + 5 min pause + 17 min light + 5 min pause + 17 min light with the applied dose of 510 J/cm^2^ (on the surface of the skull and on the area of the ethmoid bone) and 124 J/cm^2^ (on the surface of the brain) under inhalation anesthesia (1% isoflurane at 1 L/min N_2_O/O_2_—70/30 ratio).

### 2.9. Statistical Analysis

The results are presented as mean ± standard error of the mean (SEM). Differences from the initial level in the same group were evaluated by the Wilcoxon test. Intergroup differences were evaluated using the Mann–Whitney–Wilcoxon test. The significance levels were set at *p* < 0.05–0.001 for all analyses.

## 3. Results

### 3.1. The Intravasal Delivery of Liposomes to the Healthy Brain

The cribriform plate is a “window” from and into the brain as well as a connective bridge between the brain fluids and the peripheral lymphatic system [14,15,16,17,18,19]. Therefore, in the first step, confocal imaging of distribution of liposomes along the ethmoid bone and the olfactory bulb was performed in the healthy mice (Figure 1a). Figure 1b schematically demonstrates our hypothesis that after intranasal administration, liposomes with the extracellular fluids enter the lumen of LVs in the nasal mucosa and are drained in both directions, from and into the brain via the space between the OSN providing directed fluid flow [16,21,22,23,24,25]. Indeed, 3 h after intranasal administration of liposomes, we observed nanocarriers both in the brain and the meninges of mice from the HG group as well as in the brain and in the dcLNs of mice from the GBM group (Figure 1c,d,g–k and Figure 3c,d).

Notably, confocal imaging revealed directional distribution of liposomes along the main cerebral arteries (Figure 1c,d,g–i). In our previous results, we discovered that liposomes from the dcLNs enter into the MLVs with further spreading into the brain tissues [27]. In this study, we also observed the presence of liposomes in the MLVs and on the brain surface after its intranasal administration (Figure 1j,k). Based on our presented and previous data, we assume that the MLVs can play an important role in delivery of liposomes from the nasal cavity to the brain. To test this hypothesis, in the next step of the study, we analyzed the effectiveness of intranasal delivery of liposomes to the brain after photo-ablation of the MLVs.

**Figure 1 pharmaceutics-15-00036-f001:**
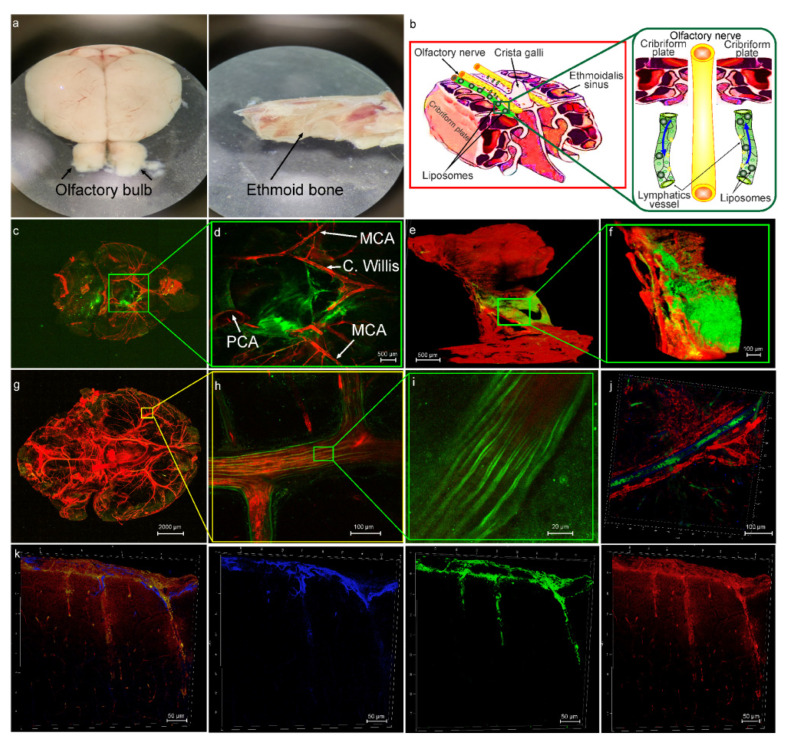
The lymphatic pathway of delivery of liposomes (green) to mouse brain from the HG group: (**a**) Images of the ethmoid bone and the olfactory bulb, which were used for confocal and multiphoton imaging of distribution of liposomes after its intranasal injection; (**b**) schematic illustration of hypothesis that after intranasal administration, liposomes with the extracellular fluids enter the lumen of LVs in the nasal mucosa and are drained in both directions, from and into the brain via the space between the OSN providing directed fluid flow; (**c**) representative confocal image of ventral part of the brain after intranasal injection of liposomes (green), the cerebral vessels were filled with the Evans Blue dye (red, i.v.); (**d**) representative confocal image of region of interest from (**c**), the image shows distribution of liposomes (green) along the main arterial vessels: MCA: middle cerebral artery, C. Willis: Circle of Willis, PCA: posterior cerebral artery; (**e**)—multiphoton image of the cribriform plate immersed in buffer solution and covered with cover glass. Objective—10 × 0.45, two photon fluorescence excitation 970 nm, emission: green (liposomes) 506—593 nm; red (Evans Blue) 604—679 nm; (**f**) multiphoton image of region of interest from (**e**), the image demonstrates distribution of liposomes (green) in the freshly removed cribriform plate stained with Evans Blue (red, emission 604—679 nm); (**g**) whole imaging of the mouse brain after intranasal delivery of liposomes; (**h**,**i**) the region of the middle cerebral artery (red, filled with Evans Blue) depicting directional distribution of liposomes; (**j**) the presence of liposomes (green) in the MLVs labeled with Live-1 (blue), the blood vessels (red) filled with Evans Blue; (**k**) representative images of distribution of liposomes (green) along the brain surface, astrocytes are labeled with GFAP (blue), the cerebral blood vessels were filled with Evans Blue (red).

### 3.2. The Role of the MLVs in Intranasal Delivery of Liposomes to the Brain

To study the role of MLVs in intranasal delivery of liposomes to the brain, we investigated the effectiveness of liposome spreading in the brain tissues after photo-impairment of the MLVs. The photo-ablation of the MLVs was performed by the visudyne injection into the cisterna magna and its excitation by 689-nm laser irradiation.

Our results revealed that 7 days after photo-ablation of the MLVs, the meningeal lymphatic coverage was significantly decreased (*p* = 0.001, the sham group vs. the laser + visudyne group; *p* = 0.001, the laser group vs. the laser + visudyne group; *p* = 0.001, the visudyne group vs. the laser + visudyne group, *n* = 5 in each group, Mann–Whitney–Wilcoxon test) (Figure 2a–e). There were no changes in the MLV network in the sham, the visudyne, or and the laser groups (Figure 2a–c,e). Thus, photo-ablation caused the loss of MLVs that is consistent with the findings presented by Chen et al. [29].

### 3.3. Intranasal Delivery of Liposomes to the Brain Is Reduced in Mice with GBM

The liposomes are promising transporters to deliver antitumoral drugs to GBM [27,43,44,45,46,47]. However, there is evidence that lymphatic drainage of the brain is impaired in animals with GBM [57,58]. To answer the question whether drainage is reduced in mice with GBM, in this study, we analyzed the distribution of liposomes in dorsal aspect of the brain and in the dcLNs, as a first anatomical station of the CSF exit, after its intranasal administration in the HG and the GBM groups.

Figure 3a–d clearly demonstrate dramatic reducing spreading of liposomes in the brain as well as their draining to the dcLNs in the GBM group vs. the HP group. Indeed, the quantitative analysis shows that the intensity of the fluorescent signal from liposomes was 2.4-fold (*p* < 0.01) higher in the brain and was 2.1-fold (*p* < 0.05) higher in the dcLNs in the HG group compared with the GBM group (Figure 3e,f). Thus, this series of experiments suggest that GBM is characterized by a significant reduction in the transport of liposomes in the nasal-brain-lymphatic system causing a decrease in the efficiency of intranasal delivery of liposomes to the brain.

### 3.4. Photostimulation of Intranasal Delivery of Liposomes to GBM

In our previous study, we discovered that photostimulation improves lymphatic delivery of liposomes from the dcLNs to the rat brain [27]. We hypothesized that photostimulation can enhance intranasal delivery of liposomes to the mouse brain. To test our hypothesis, we performed the confocal analysis of the accumulation of liposomes in GBM after its intranasal administration in mice treated and not treated by laser irradiation. Figure 4a,b show that the GBM + laser group demonstrates higher accumulation of liposomes in the GBM tissues than the GBM group without laser treatment. So, the intensity of signal from liposomes in GBM was 2.9-fold higher (*p* < 0.01) in mice treated by laser compared with intact mice (Figure 4c). These results suggest that photosimulation increases the efficiency of intranasal delivery of liposomes to GBM, presumably due to photo-modulation of lymphatic transport of nanocarriers to the brain tissues [27,28] (Figure 4d).

## 4. Discussion

In this study on mice, we clearly demonstrate intranasal delivery of the GM_1_-liposomes into the mouse brain and GBM via the nasal-brain-lymphatic system, bypassing the BBB. Our data are consistent with the results of other studies suggesting that intranasal administration of therapeutics is an effective method for brain drug delivery [9,10,11,12,13]. In the experiments with ablation of the MLVs, we report an important role of the meningeal “door” for transport of liposomes from the nasal cavity to the brain tissues.

GM_1_—the fundamental lipid component of the outer leaflets of mammalian cell membranes especially abundant on the plasma membranes of neurons. The neurotropic and neuroprotective properties of GM_1_ have long been known (for example, reviewed in [59]). Therefore, as such, its inclusion in the formulation for the treatment of brain diseases may have a beneficial effect. But an even more important reason is that the GM_1_ liposomes have shown good stability in biological environments [60,61,62] and a suitable bio-distribution for delivery of the cargo to the brain [63]. The delivery of GM_1_ liposomes to the brain from the bloodstream [54] and from the lymph [27] has been shown. At the dawn of research on the development of stealth liposomes that are able to avoid premature withdrawal by the immune cells, it was shown that glycolipids, and first of all GM_1_, prolong circulation half-life of nano-sized liposomes [60,61]. Presumably, this effect is due to shielding of the liposome surface by voluminous and highly hydrated carbohydrate head groups. Later, instead of glycolipids, less expensive lipid conjugates with polyethylene glycol evolved as protecting components for liposomal formulations. Although, well-known pegylation of drug delivery systems often lead to problems associated with immunity against pegylation (reviewed in [64]).

The cribriform plate is the center of lymphatic pathway of metabolic clearance from the brain [14,15,16,17,18,19]. This plate has a fenestrated anatomical structure of the ethmoid bone, which is rich in the LVs. The CSF flowing through the cribriform plate is absorbed by the LVs in the nasal mucosa and is drained in both directions, to the peripheral dcLNs (removal of metabolites from the brain) and to the brain (immune communications) [25,26,27,28,29,30,65,66]. Thus, the lymphatic system of the cribriform plate is a door from and into the brain, which maintains balance between the exit and the entrance of molecules and cells from and into the CNS [14,15,16,17,18,19]. We assume that the GM_1_-liposomes enter the lumen of the LVs located in the cribriform plate after its administration in the nasal cavity and then are drained with the lymph to the brain tissues reaching GBM as we observed in our studies.

We revealed that GBM is characterized by a dramatically reduced intranasal delivery of liposomes to the brain. This can be explained by a decrease in the MLVs network and the reduced outflow of the CSF in mice with GBM that was reported in other investigations [63,64,65,66,67]. Hu et al. demonstrate that the dorsal MLVs undergo extensive remodeling in mice with GL261 glioma that is accompanied by changes in the gene controlling lymphatic remodeling, fluid drainage, inflammatory, and immunological responses [57]. Ma et al. documented significantly reduced lymphatic drainage of the brain tissues in mice with GBM [58].

Our results clearly show that infra-red photostimulation (1267 nm) significantly improves intranasal delivery of liposomes to the brain and to GBM. The infra-red light of 800–1100 nm is widely used for the photo-therapy of brain diseases [31,32,33,34,35,36,37]. However, infra-red phototherapy has significant limitations, such as low penetration into the brain due to light scattering and heating effects [68]. The light wavelength of 1300 nm has less scattering and can penetrate deeper into the brain [69]. Therefore, we selected laser wavelength (1267 nm) with the dose 510 J/cm^2^ (on surface of the skull and the ethmoid bone) and 124 J/cm^2^ (on the surface of the brain) that we estimated as the effective irradiation dose for photo-stimulation of the MLVs and lymphatic delivery of liposomes to GBM in our previous study [27].

To study the mechanisms of the therapeutic effects of photostimulation using infra-red laser irradiation (1267 nm), in our recent study, we reported photo-influences on the diameter of the basal MLVs and the lymphatic contractility as well as on the nitric oxide (NO)-mediated regulation of the MLVs tone [27,70]. Our previous findings demonstrate that photostimulation dilates the MLVs and increases the NO production in the lymphatic endothelium associated with an increase in contractility of the LVs. We clearly show that photo-mediated activation of the lymphatic contractility and the NO synthesis in the lymphatic endothelium might be the possible mechanisms responsible for the photo-stimulation of the lymphatic transport of compounds from and into the brain [27,70].

The photo-mediation of the lymphatic transport of different compounds can be explained by a photo-related increase in the activity of the endothelial NO synthase in the lymphatic endothelium [71]. The NO acts as vasodilator via the opening of calcium-activated potassium channels and reuptake of Ca^2+^ leading to relaxation of the LVs [72]. There are several other mechanisms responsible for the NO-mediated regulation of relaxation and contractility of the LVs: (1) The activation of iron-dependent enzymes, including mitochondrial acontiase, an [Fe-S] protein in macrophages [73]; (2) inactivation of ribonucleotide reductase [74] and aconitase [75]; the stimulation of synthesis of the ADP-ribosylation of glyceraldehyde-3-phosphate dehydrogenase [76] and protein-sulfhydryl-group nitrosylation [77]. In our previous work, we demonstrated that photostimulation induces relaxation of the mesenteric LVs and causes a decrease in the expression of the tight junction proteins proving an increase in the permeability of LVs [39]. We assume that photostimulation of the LVs located in the cribriform plate and the MLVs facilitates lymphatic transport of liposomes to the brain tissues and to GBM. Our findings are consistent with our previous data demonstrating photostimulation of lymphatic delivery of liposomes to GBM from the dcLNs [27]. Further studies of photo-effects on vascular permeability using proven methods [78,79,80,81,82] will advance a better understanding of the mechanisms of photostimulation of lymphatic drug delivery into the brain.

Photostimulation has been shown to produce highly pleiotropic biological effects (other than lymphatic engagement) [32,37]. The well-studied mechanism of photostimulation is focused on the mitochondrial cytochrome c oxidase (CCO), which is responsible for the final reduction of oxygen to water using the electrons generated from glucose metabolism [83]. Photostimulation causes an increase in production of NO, which inhibits the CCO enzyme activity that is accompanied by an increases of mitochondrial membrane potential contributing more oxygen consumption, more glucose metabolization, and more ATP production by the mitochondria [84,85,86,87,88,89]. There is evidence that photostimulation activates generation of the reactive oxygen species (ROS) in the mitochondria. The ROS trigger different mitochondrial signaling mechanisms leading to cytoprotective, antioxidant, and anti-apoptotic effects in the cells [85].

## 5. Conclusions

In summary, our findings clearly demonstrate that intranasal administration of liposomes is accompanied by delivery of nanocarriers to the mouse brain reaching GBM via the lymphatic pathway, in which the MLVs play an important role. The GBM is characterized by dramatic reduction in transport of liposomes via the nasal-brain-lymphatic system. Photostimulation significantly improves intranasal delivery of liposomes to GBM, presumably due to photo-modulation of the lymphatic transport of liposomes from the nasal cavity to the brain tissues reaching GBM. Thus, these results open new perspectives for non-invasive improvement of the efficiency of intranasal delivery of anti-tumor drugs to the brain tissues using nanocarriers and infra-red laser-based technologies. We would like to emphasize that we present data that require further detailed experimental study examining the sufficiency of the proposed method to generate an appropriate therapeutic effect of cancer drugs. Photobiomodulation came as a therapeutic technique in the 1960s [37]. Currently, there are numerous commercial laser-based therapeutic devices that are widely used in clinics [37]. Therefore, photostimulation of intranasal delivery of liposomes to the brain and to the GBM tissues has great prospects to be quickly introduced into clinical practice and to be an innovative strategy in overcoming the obstacles facing nose-to-brain delivery of promising therapeutics.

## Figures and Tables

**Figure 2 pharmaceutics-15-00036-f002:**
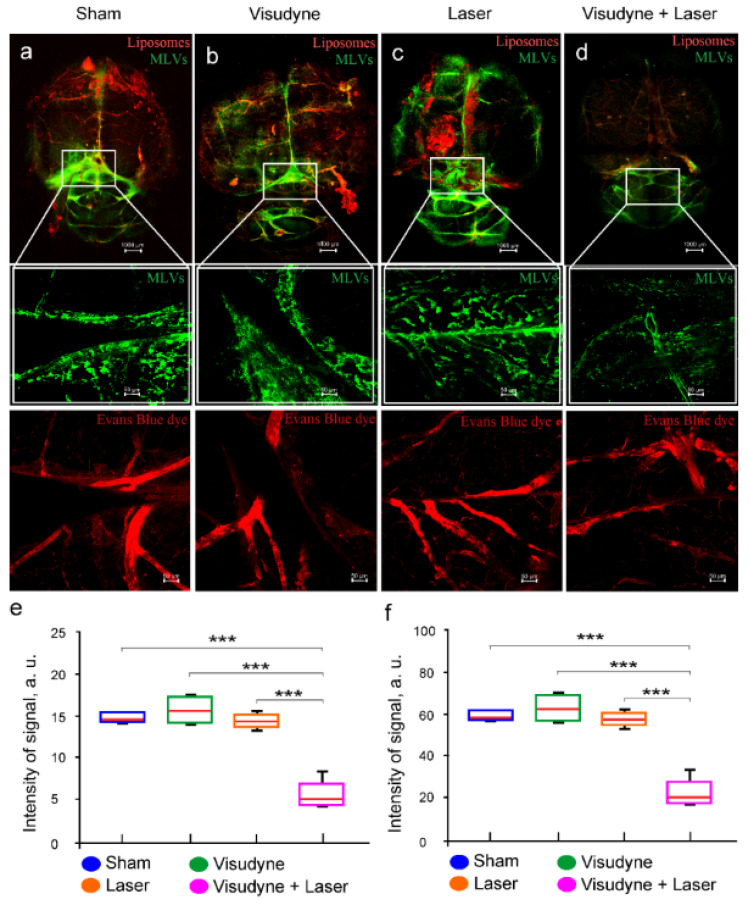
The effects of photo-ablation of the MLVs (green) on intranasal delivery of liposomes (ex.578/em.602) to the brain. (**a**–**c**) Representative confocal images of the MLVs labeled with Lyve-1 (green) in the sham group (**a**), in the visudyne group (**b**), in the laser group (**c**), and in the visudyne + laser group (photo-ablation of the MLVs) (**d**), *n* = 5 in each group. The cerebral vessels were filled with Evans Blue (ex.640/em.680); (**e**,**f**) quantification of percentage area of the LYVE-1-positive MLVs (**e**) and the intensity of fluorescent signal from liposomes (**f**) in the brain tissues in the tested groups, *n* = 5 in each group, *** *p* < 0.001 the Mann–Whitney–Wilcoxon test.

**Figure 3 pharmaceutics-15-00036-f003:**
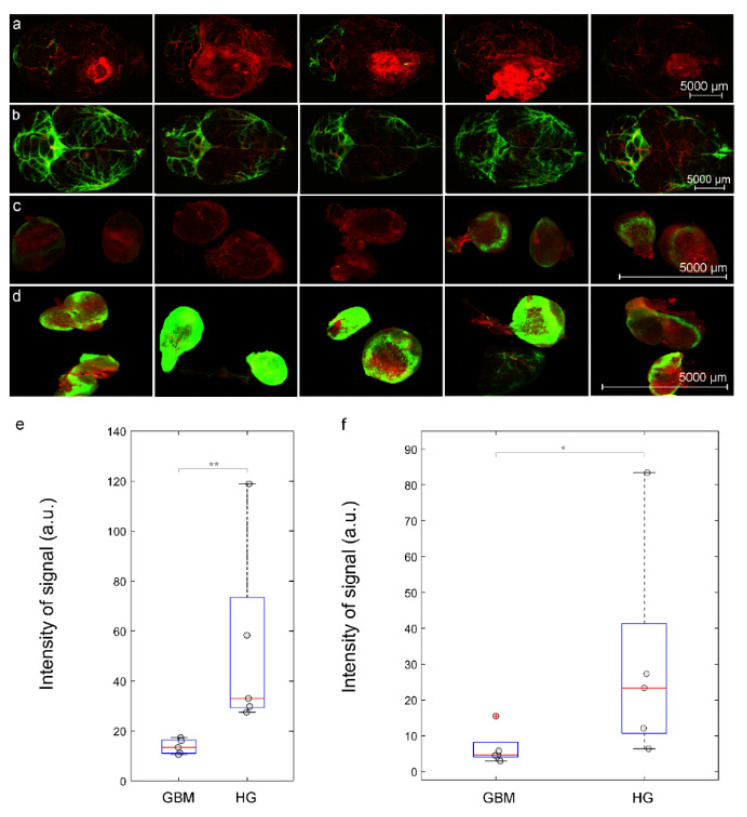
The GBM (red, λ_ex_ = 555 nm) is characterized by reducing intranasal delivery of liposomes to the brain and their removal to the dcLNs: (**a**)–(**d**) representative confocal images of liposomes (green) spreading in dorsal part of the brain and their accumulation in the dcLN in the GBM group (**a**,**c**) and in the HP (**b**,**d**), the blood vessels were filled with Evans Blue (red, λ_ex_ = 640 nm); the quantitative analysis of fluorescent signal from liposomes (a.u.) in the brain (**e**) and in the dcLNs (**f**) in the GBM group and in the HG groups, *n* = 5 in each group, * *p* < 0.05, and ** *p* < 0.01 the Mann–Whitney–Wilcoxon test, the outliers of experimental data are plotted using the + symbol.

**Figure 4 pharmaceutics-15-00036-f004:**
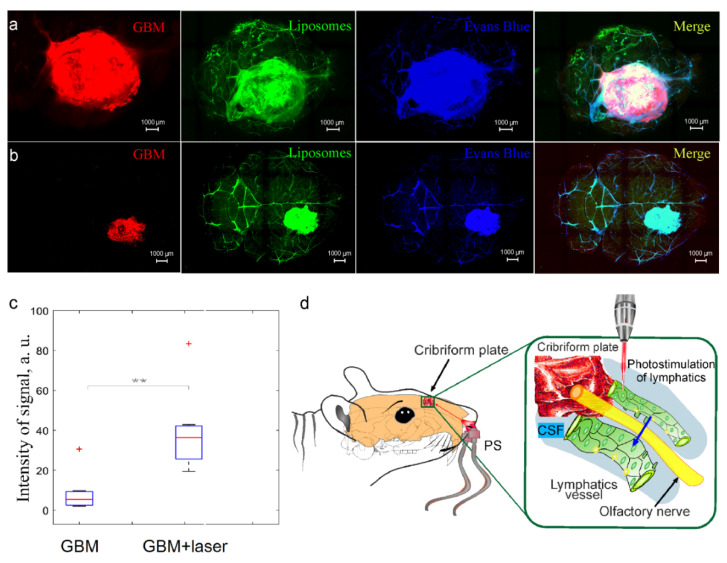
(**d**) Photostimulation of intranasal delivery of liposomes to GBM: (**a**,**b**) Representative images of the mouse brain with GBM (red, Ex 561/Em 570–620) 3 h after intranasal administration of liposomes (green, Ex 488/Em 500–550) in mice not treated (**a**) and treated by laser (**b**), the blood vessels were filled with the Evans Blue dye (blue, Ex 640/Em 663–738); (**c**) the quantitative analysis of fluorescent signal from liposomes (a.u.) in the GBM tissues in the GBM + laser and the GBM groups, *n* = 7 in each group, ** *p* < 0.01, the Mann–Whitney–Wilcoxon test, the outliers of experimental data are plotted using the ‘+’ symbol.

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
