# Peer review of "Intranasal Delivery of Liposomes to Glioblastoma by Photostimulation of the Lymphatic System"

_pharmaceutics, 2022, doi:10.3390/pharmaceutics15010036_

Round 1

Reviewer 1 Report

The authors have almost identically repeated their work which they cited previously, to the point that the methodologies section is near identical (while changed slightly just to make it not copy pasted). They repeated the exact same experiments and designs using a different cell line and claiming the importance of the cribiform plate. Yet in the conclusion they don't use that term but use MLV, which had already been studied in the previous paper. They even go as far as to call this Pilot study data which is the second article of the same design. If there is novelty in this paper the authors did not show it in their explanation of the difference between the previously published work and this submission

Author Response

The authors express their deep gratitude for reviewing our article and critical comments, which are very important to improve the quality of our manuscript.

Indeed, two pilot papers (Nanophoton 2021, 10(12), 3215-3227; Nano Lett 2020, 20(7), 5415-5420) have been published showing for the first time that it is possible to inject compounds and liposomes into the deep cervical lymph nodes (dcLNs) and deliver them to brain tissue and glioma in rats. These works were not aimed at the clinical significance due to the anatomical features of the deep location of dcLNs, which requires surgical intervention for the introduction of drugs into them. However, this approach was justified by the fact that the direct injection of dyes and liposomes into the dcLNs with subsequent monitoring of their transport to the brain tissues makes it possible to conclude that the lymphatic system plays an important role in the delivery of tracers to the central nervous system. Thus, the above works were just an important informative and fundamental platform for the further development of methods for the lymphatic delivery of drugs to the brain tissues, bypassing the blood-brain barrier, which can be introduced into clinical practice.

Our studies have shown for the first time that the introduction of liposomes into the nose is accompanied by their distribution in the brain tissue, and is more effective after photostimulation. Using the original protocol, we have shown that photoablation (experiments with Visydine) of the meningeal lymphatic vessels significantly reduces the intranasal delivery of liposomes to the brain. Thus, the important role of meningeal lymphatics as a lymphatic connecting pathway between the lymphatic vessels of the nose and the meninges was shown for the first time. The decreased lymphatic drainage in mice with GBM and possibility of intranasal delivery of liposomes to GBM as well as to the brain tissues was also shown for the first time, which has an important clinical significance. Notice that in most cases intranasal delivery of drugs is not sufficient to achieve a therapeutic effect due to brain drug delivery in a short distance. Therefore, our results open new perspectives for non-invasive improvement of efficiency of intranasal delivery of cancer drugs to the brain tissues using nanocarries and near-infrared laser-based therapeutic devices, which are commercially available and widely used in clinical practice. Currently, there are numerous commercial laser-based therapeutic devices that are widely used in clinics. Therefore, photostimulation of intranasal delivery of liposomes to the brain and to the GBM tissues has great prospects to be quickly introduced into clinical practice and to be an innovative strategy in overcoming the obstacles facing nose-to-brain delivery of promising therapeutics. 

It is important to note that in the previous publication we used only transcranial photostimulation with a dose 1611 J/cm2 on the surface of skull and 470 J/cm2 on the surface of the brain, in this study, photostimulation was performed both through the nose (510 J/cm2) and transcranially (510 J/cm2 on the surface of skull and 124 J/cm2 on the surface of the brain) (Lines 257-270 and 431-440).

Another important aspect is that our studies are performed on mice with the cell line EPNT-5 of mouse GBM, which differs significantly from our previous study performed on rats with C6 glioma cell line. The selection of mice with the cell line EPNT-5 of GBM was not random. This was due to the fact that C6 arises from chemical mutagenesis in rats, and many investigators would consider C6 as a poor model for human GBMs. Model C6 is used mainly for basic research that is not aimed at clinical application. The cell line EPNT-5 of GBM has certain metabolic and structural characteristics close to those of a human GBM, in particular hyper-expression of α-glycerophosphate dehydrogenase enzyme and GABA-A channel subunit, cellular polymorphism, high genetic heterogeneity, which makes GBM resistant to therapy (Kondakova L et al. The approach for using the transplatable tumorous brain cell lines in experimental neuromorphology and neuropathology. Clinical and Experimental Morphology. 2012 (2). http://en.morfolhum.ru/journal/2012-2).

We added these explanations in the manuscript (Lines 63-72) and in the methods (153-157).

Repetitions in the methods of our article occur only in those places where the synthesis of liposomes, the established research protocols and the statistical analysis are described.

Thus, the idea, results, methods (GBM model, ICH, confocal and multiphoton approaches, modulations of the meningeal lymphatic system), conclusions, and especially the clinical significance of our studies are fundamentally different from our previous publication.

The authors would like to thank the referee again for the great help in improving our paper for its possible publication in Pharmaceutics.

Reviewer 2 Report

This paper introduces a method to deliver liposome to glioclastoma using photostimulation. As they mentioned in line 72, liposomes are promising transporters. In addition, the topic of drug delivery to the mouse brain bypassing BBB is very important. Overall, the manuscript is a good one. To highlight the importance, I recommend the following points to be addressed before the publication:

1.       To research topic is important, but the novelty of this research is not clearly delivered. As mentioned in the paragraph line 63, regulation of permeability using photostimulation was already developed and reported several times. Liposome delivery using photostimulation was also reported. The main point should be GM1 liposome delivery, but the characteristics of GM1 were not clearly explained. Therefore, the reasoning in discussion is not also clear enough. To make the point clear, structural or chemical property of GM1 should be explained, and the property needs to be connected to discussion.

2.       In line 436, they mentioned “we present pilot data”. Considering the similar studies have been published already, the pilot data is not enough to emphasize the novelty of this research. If the detailed experimental study and analysis are clearly emphasized, the novelty points can be much more clearly delivered. The detailed discussion can be based on previously reported papers about optical imaging methods on BBB (1, 2) and quantification method of permeability and effects (3-5)

3.       In line 316, the authors mentioned “The liposomes are promising transporters to deliver antitumoral drugs to GBM X[27, 43-47]”. The reason why liposomes are promising transporters is important for this paper, but, by only citing related papers and did not mention anything about the rationales. This lack of information makes readers hard to understand this important point, so I recommend summarize this information in the text for readers.

1.          Shi L, Sordillo LA, Rodríguez-Contreras A, Alfano R. Transmission in near-infrared optical windows for deep brain imaging. J Biophotonics. 2016;9(1-2):38-43.

2.            Wei W, Wang X, Li Y, Cheng Y, Fung AA, Yang X, et al. Chapter Five - Advances in optical imaging of drug delivery across the blood-brain barrier. In: Visser TD, editor. Progress in Optics. 66: Elsevier; 2021. p. 171-253.

3.            Shi L, Palacio-Mancheno P, Badami J, Shin DW, Zeng M, Cardoso L, et al. Quantification of transient increase of the blood-brain barrier permeability to macromolecules by optimized focused ultrasound combined with microbubbles. Int J Nanomedicine. 2014;9:4437-48.

4.            Shi L, Zeng M, Fu BM. Temporal effects of vascular endothelial growth factor and 3,5-cyclic monophosphate on blood-brain barrier solute permeability in vivo. J Neurosci Res. 2014;92(12):1678-89.

5.            Shi L, Zeng M, Sun Y, Fu BM. Quantification of blood-brain barrier solute permeability and brain transport by multiphoton microscopy. J Biomech Eng. 2014;136(3):031005. 

Author Response

The authors would like to express their sincere gratitude to the referee for valuable advices and recommendations, which significantly improve the quality of our article.

We added in the discussion the characteristics of liposomes that make them attractive for drug delivery to the brain tissue (Lines 397-412, Ref. 57-62, Lines 616-629) as well as we cited the recommended articles (Ref. 78-82, Lines 696-706).

Regarding the novelty of our research, we have added an explanation to the introduction (Lines 63-72) and in the methods (Lines 153-157).

Indeed, two pilot papers (Nanophoton 2021, 10(12), 3215-3227; Nano Lett 2020, 20(7), 5415-5420) have been published showing for the first time that it is possible to inject compounds and liposomes into the deep cervical lymph nodes (dcLNs) and deliver them to brain tissue and glioma in rats. These works were not aimed at the clinical significance of the fundamental results due to the anatomical features of the deep location of dcLNs, which requires surgical intervention for the introduction of drugs into them. However, this approach was justified by the fact that the direct injection of dyes and liposomes into dcLNs with subsequent monitoring of their transport to the brain tissues makes it possible to conclude that the lymphatic system plays an important role in the delivery of tracers to the central nervous system. Thus, the above works were just an important informative and fundamental platform for the further development of methods for the lymphatic delivery of drugs to the brain tissues, bypassing the blood-brain barrier, which can be introduced into clinical practice.

Our studies have shown for the first time that the introduction of liposomes into the nose is accompanied by their distribution in the brain tissue, and is more effective after photostimulation. Using the original protocol, we have shown that photoablation (experiments with visydine) of the meningeal lymphatic vessels significantly reduces the intranasal delivery of liposomes to the brain. Thus, the important role of meningeal lymphatics as a lymphatic connecting pathway between the lymphatic vessels of the nose and the meninges was shown for the first time. The decreased lymphatic drainage in mice with GBM and possibility of intranasal delivery of liposomes to GBM and the brain tissues was also shown for the first time, which has an important clinical significance. Notice that in most cases intranasal delivery of drugs is not sufficient to achieve a therapeutic effect due to brain drug delivery in a short distance. Therefore, our results open new perspectives for non-invasive improvement of efficiency of intranasal delivery of cancer drugs to the brain tissues using nanocarries and near-infrared laser-based therapeutic devices, which are commercially available and widely used in clinical practice. Currently, there are numerous commercial laser-based therapeutic devices that are widely used in clinics. Therefore, photostimulation of intranasal delivery of liposomes to the brain and to the GBM tissues has great prospects to be quickly introduced into clinical practice and to be an innovative strategy in overcoming the obstacles facing nose-to-brain delivery of promising therapeutics.

It is important to note that in the previous publication we used transcranial photostimulation with a dose (1611 J/cm2 on the surface of skull and 470 J/cm2 on the surface of the brain), in this study, photostimulation was performed both through the nose with a low laser intensity (510 J/cm2) and transcranially (510 J/cm2 on the surface of skull and 124 J/cm2 on the surface of the brain).

Another important aspect is that our studies are performed on mice with the cell line EPNT-5 of mouse GBM, which differs significantly from our previous study performed on rats with C6 glioma cell line. The selection of mice with the cell line EPNT-5 of GBM was not random. This was due to the fact that C6 arises from chemical mutagenesis in rats, and many investigators would consider C6 as a poor model for human GBMs. Model C6 is used mainly for basic research that is not aimed at clinical application. The cell line EPNT-5 of GBM has certain metabolic and structural characteristics close to those of a human GBM, in particular hyper-expression of α-glycerophosphate dehydrogenase enzyme and GABA-A channel subunit, cellular polymorphism, high genetic heterogeneity, which makes GBM resistant to therapy (Kondakova L et al. The approach for using the transplatable tumorous brain cell lines in experimental neuromorphology and neuropathology. Clinical and Experimental Morphology. 2012 (2). http://en.morfolhum.ru/journal/2012-2).

Repetitions in the methods of our article occur only in those places where the synthesis of liposomes or established research protocols and statistical analysis are described.

Thus, the idea, results, methods (GBM model, ICH, confocal and multiphoton approaches, modulations of the meningeal lymphatic system) conclusions, and especially the clinical significance of our studies are fundamentally different from our previous publication.

The authors once again thank the referee for great help in improving our paper for its possible publication in Pharmaceutics.

Reviewer 3 Report

The article submitted to Pharmaceutics with the title “Photostimulation increases intranasal liposome delivery to 2 glioblastoma via the lymphatic route” tries to demonstrate the contributions of lymphatic vessels crossing the cribriform plate to the nose-to-brain delivery of liposomes and describes the effect that near-infrared photostimulation of the area of the cribriform plate and the meninges had in labelled liposomes distribution to glioblastoma in mice.  This effect was assessed by fluorescence microscopy, by using fluorescently labelled liposomes composed of phospholipid and ganglioside GM1.

Both the hypothesis of nose-to-brain distribution through the lymphatic route and a possible enhancing effect of light irradiation to increase brain delivery are not commonly addressed and deserve attention. However, several points require clarification.

1. Were controls performed to make sure that the fluorescence was indeed due to what is said to be? For example, were images collected in tissues without liposomes to compare to the level of autofluorescence in that particular wavelength? Also, the methods presentation is confusing…does the section 2.6 corresponds to acquisition of imagens of the tissue section prepared and stained as in section 2.4?

2. When and how was Evans Blue used? Does it mask green fluorescence present in the blood vessels?

3. How is the discrete (stronger green points) of green fluorescence interpreted vs the diffuse background in Figure 1a? it is also very difficult to identify the underlying structure… is there bone present? Was the tissue not decalcified? (it was not mentioned, nor the plane of tissue sectioning). Labeling cells or nuclei, or other structures present may also help to give references for interpretation.

4. I do not understand how “directional movement of liposomes” or any movement can be seen in fixed tissue and I am not able to interpret the green “stripe” like structures. What do you mean by directional movement?

5. Does the characteristic of negatively charge liposomes or specifically the ganglioside GM1 contribute to pattern of liposomal distribution, or would it be observed with liposomes of different composition? Are they expected to diffuse or be taken up by immune cells and then be transported?

6. 3h after administration it could be the elimination phase already – why did you choose 3h and why is it interpreted that liposome (or the label) is going into the brain and not out of the brain through lymphatic vessels?

7. You mention “the blood vessels are 200 not presented in Figure 2 and are available in data availability statement” – I do believe that it should be shown that the blood vessels were not compromised by Visudyne treatment.

8. Figure 2 corresponds to how long after intranasal administration of liposomes?

9. Is figure 3 showing dorsal or ventral (as said in the text) part of brains? Is it a slice? Was Evans Blue also used in Figure 3 a and c? Correct HP and NG…

10. In figure 4 has the glioblastoma spread? (there is red spread around…)

11. Also in figure 4, do not the liposomes coincide with Evans Blue this time?

12. In interpretation of data related to Figure 4, the hypothesis was increased delivery to the brain, but the quantification was of tumor mass, increased delivery to healthy brain tissue was not shown.

13. Also, could the irradiation do something to the tumor mass itself? Or increase delivery through the blood compartment?

Author Response

The authors would like to thank the referee for constructive advices and important comments.

In figure (in the attached file) we show confocal images of the mouse brains without injection of liposomes. The brains were taken after intravenous administration of Evans Blue. No liposomes were injected. These control images were taken with confocal microscope settings identical to that used for Figure 3 brain images. Maximum intensity projection was calculated over 5 slices captured with 500 um z step, objective 2x0.1, pinhole size 83 um,  images were stitched of 4×3 fields of view. Left panels corresponds to output lookup same as for fig 3 post processing. Right panel green lookup min = 0, max = 34, so the backgroung PMT noise is visible in green channel. It is clear that green autofluorescence is at noise level, while the green fluorescence of liposomes shown on fig.3 is at least one order of magnitude higher.

Comment: Also, the methods presentation is confusing…does the section 2.6 corresponds to acquisition of images of the tissue section prepared and stained as in section 2.4?

Response: We made correction the sections 2.4 and 2.6 (Lines 183-238).

Comment: When and how was Evans Blue used? Does it mask green fluorescence present in the blood vessels?

Response: We added in the methods description when and how we made Evans Blue injection (Lines 224-226).

Evans Blue dye (T-1824) is a well-known tag for blood plasma proteins [American Journal of Physiology-Legacy Content 151.1 (1947): 26-33, https://doi.org/10.1152/ajplegacy.1947.151.1.26]. It was administered intravenously. Complexes of Evans blue dye with blood plasma proteins are fluorescent (peak of excitation at 620 and emission at 680 nm) [Journal of neuroscience methods 8.1 (1983): 41-49., https://doi.org/10.1016/0165-0270(83)90050-X]. Evans Blue dye was imaged in far red channel (ex. 640 nm, em. 663-739) using Nikon A1R MP of DU4 confocal detector. Evans Blue dye was used as specific fluorescent marker of blood plasma proteins but not as a mask agent to suppress of autofluorescence [Histochemistry 82.3 (1985): 205-208. https://doi.org/10.1007/BF00501396].

Comments: How is the discrete (stronger green points) of green fluorescence interpreted vs the diffuse background in Figure 1a? it is also very difficult to identify the underlying structure… is there bone present? Was the tissue not decalcified? (it was not mentioned, nor the plane of tissue sectioning). Labeling cells or nuclei, or other structures present may also help to give references for interpretation.

Response: We changed Figure 1a to demonstrate clearer distribution of liposomes along the ethmoid bone and the olfactory bulb. Fresh samples of whole brain and the ethmoid bone were immersed in saline solution in a Petri dish and covered with horizontally oriented cover glass. The Petri dish was placed on an electromechanical stage of the confocal imaging system. The confocal imaging of distribution of liposomes in the ethmoid bone, the olfactory bulb, the meninges and the brains was taken 3 hours after its intranasal delivery. We added this description in the methods (Lines 208-229).

Comments: I do not understand how “directional movement of liposomes” or any movement can be seen in fixed tissue and I am not able to interpret the green “stripe” like structures. What do you mean by directional movement?

Response: We apologize for the inaccuracy in this statement. Figures 1 h and i illustrate directional distribution of liposomes.

Comments: Does the characteristic of negatively charge liposomes or specifically the ganglioside GM1 contribute to pattern of liposomal distribution, or would it be observed with liposomes of different composition? Are they expected to diffuse or be taken up by immune cells and then be transported?

Response: We have introduced into the discussion a paragraph devoted to the role of GM1 in liposomes (Lines 398-414). We assume the choice of this glycolipid for our formulation is rational and justified. It is difficult to imagine what effect liposomes of a different composition would have. Besides, relatively recently it has been shown that the oligosaccharide chain of the molecule is responsible for bioactive properties of GM1 (Mol Neurobiol. 2019, 56(10):6673-6702. doi: 10.1007/s12035-019-1556-8). As far as we know negative charge as such does not prevent the uptake of liposomes by monocytes. Concerning the mechanism of liposome transportation to the brain, it could be the subject of our further research.

Comment: 3h after administration it could be the elimination phase already – why did you choose 3h and why is it interpreted that liposome (or the label) is going into the brain and not out of the brain through lymphatic vessels?

Response: 3 hours for confocal imaging of distribution of liposomes in the tested tissues were chosen due to the fact that this is the optimal time for a sufficient accumulation of a fluorescent signal from liposomes for optical detection. Indeed, there is both direction of the lymphatic flow, retrograde to the CNS and out of the brain tissue. Since, after intranasal injection of liposomes, they effectively accumulated in GBM and distributed in the brain tissues, especially after photoexposure, it can be assumed that the retrograde lymphatic transport of liposomes is sufficient for delivery of liposomes into the CNS.

Comment: You mention “the blood vessels are not presented in Figure 2 and are available in data availability statement” – I do believe that it should be shown that the blood vessels were not compromised by Visudyne treatment.

Response: We added in Figure 2 confocal images of the cerebral vessels filled with Evans Blue.

Comment: Figure 2 corresponds to how long after intranasal administration of liposomes?

Response: Liposomes were injected into the nasal cavity and allowed to be distributed in the tested tissues during 3 hours. Afterward, mice were decapitated, the meninges were removed and prepared for confocal microscopy presented in Figure 2. We added in the methods description of this protocol (Lines 224-226).

Comments: Is figure 3 showing dorsal or ventral (as said in the text) part of brains? Is it a slice? Was Evans Blue also used in Figure 3 a and c? Correct HP and NG…

Response: We apologize for the inaccuracies in the text. Figure 3 shows dorsal part of the brain where GBM was most visible. We made a correction in Figure 3 as well as we changed NG to HG.

Figure 3 presented fresh whole brain quickly removed from mice and immediately analyzed using confocal microscopy. Evans Blue dye was injected 30 min before brain removing and further confocal imaging. We added this description in the text of manuscript (Lines 215-218 and 224-226).

Comments: In figure 4 has the glioblastoma spread? (there is red spread around…)

Response: In process of figure 4 preparation output lookup was accidentally oversaturated, so the GBM appears without details and background autofluorescence of tissues and the blood vessels is enhanced. We apologize for the inaccuracy. Since the original confocal images were captured without saturation with 12 bit per pixel in each channel, we revised fig. 4 to avoid oversaturation with output RGB lookup. Substantial autofluorescence of tissues always presents on confocal images at 488 nm and 561 nm. Spread in GBM channel appears as a result of superposition of brain shell vessels autofluorescence in max intensity projection of 5 confocal slices. In contrast to GBM the artifact is visible only in 561 nm channel, when the GBM appears in all channels, because there are blood and liposomes in tumor. We changed Figure 4.

Comment: Also in figure 4, do not the liposomes coincide with Evans Blue this time?

Response: This was due to autofluorecence of the blood vessels, which were visible in 488 nm channel (green).

Comment: In interpretation of data related to Figure 4, the hypothesis was increased delivery to the brain, but the quantification was of tumor mass, increased delivery to healthy brain tissue was not shown.

Response: The results of distribution of liposomes to the brain of mice from the HG group (control) are presented in Figure 1 and Figure 3 b and c. We made correction in a caption of Figure 1 as well as in the text of manuscript (Lines 279-289, Line 301, Lines 325-344)

Comment: Also, could the irradiation do something to the tumor mass itself? Or increase delivery through the blood compartment?

Response: We discussed above about the role of blood compartment in intranasal delivery of liposomes into the brain. Regarding the impact of laser irradiation directly on GBT, since the irradiation is carried out through the nose and in a low intensity mode, such an impact is unlikely. We have unpublished results, which are now under peer review in Neuro Oncology showing that a transcranial 4 weeks-course of laser therapy (1267nm, 12.6 kJ/cm2) suppresses the GBT progression and increases the BBB permeability. However, these effects are manifested during the 4 weeks-course laser exposure through the skull to the tumor in a high irradiation dose. Therefore, it cannot be compared with the results presented in our article, where low irradiation dose was used and through the nose.

The authors would like to thank the referee again for great help in improving our paper for its possible publication in Pharmaceutics.

Reviewer 4 Report

The article is well designed and clearly presents the results. As a minor issue, I missed two things in the introduction: the visual presentation of the photostimulation therapy, the schematic presentation of the delivery to the target and the specific release of the API (visudyne); and the mechanistic presentation of the API (structure of API and its effect and photoactivation from the liposomal formulation).

Apart from these minor comments, I support the acceptance of the announcement for publication.

Author Response

The authors would like to express their deep gratitude for the review of our article and important comments for its improvement.

We apologize for the possibly unclear presentation of the results in the text of the article. Visudyne has been used as a method for photo-damage of the meningeal lymphatic vessels to study their role in liposome transport. We did not use Visudyne for photostimulation of liposome delivery to the brain tissue and we did not investigate the target and the specific release of the Visudyne. Visudyne was used according to the published protocol (Nat Commun, 2020, 11, 3159. https://doi.org/10.1038/s41467-020-16851-zl; Cell Res, 2020, 30(3):229-243. doi: 10.1038/s41422-020-0287-8) to reduce the density of lymphatic vessels in the meninges. To do this, Visudyne was introduced into the cisterna magna 7 days before the experiment. Afterward, the mice were injected with liposomes into the nose for the study of their distribution in the meninges and in the brain tissues. We improved the description of Visudyne-mediated damages of the meningeal lymphatic vessels with the aim to study of their role in transport of liposomes to brain tissues (Lines 241-253).

The authors would like to thank the referee again for the great help in improving our paper for its possible publication in Pharmaceutics.

Round 2

Reviewer 3 Report

The authors have made sufficient corrections and clarifications.

Regards